# Flaxseed Gum Solution Functional Properties

**DOI:** 10.3390/foods9050681

**Published:** 2020-05-25

**Authors:** Yingxue Hu, Youn Young Shim, Martin J.T. Reaney

**Affiliations:** 1Department of Food and Bioproduct Sciences, University of Saskatchewan, Saskatoon, SK S7N 5A8, Canada; yih018@mail.usask.ca; 2Department of Plant Sciences, University of Saskatchewan, Saskatoon, SK S7N 5A8, Canada; 3Prairie Tide Diversified Inc., Saskatoon, SK S7J 0R1, Canada; 4Guangdong Saskatchewan Oilseed Joint Laboratory, Department of Food Science and Engineering, Jinan University, Guangzhou 510632, China; 5Department of Integrative Biotechnology, College of Biotechnology and Bioengineering, Sungkyunkwan University, Suwon, Gyeonggi-do 16419, Korea

**Keywords:** *Linum usitatissimum* L., flaxseed, gum, foamability, emulsification, freeze-thaw

## Abstract

Flaxseed gum (FG) is a by-product of flax (*Linum usitatissimum* L.) meal production that is useful as a food thickener, emulsifier, and foaming agent. FG is typically recovered by hot-water extraction from flaxseed hull or whole seed. However, FG includes complex polymer structures that contain bioactive compounds. Therefore, extraction temperature can play an important role in determining its functional properties, solution appearance, and solution stability during storage. These characteristics of FG, including FG quality, determine its commercial value and utility. In this study, FG solution functional properties and storage stability were investigated for solutions prepared at 70 and 98 °C. Solutions of FG prepared at 98 °C had lower initial viscosity than solutions extracted at 70 °C; though the viscosity of these solutions was more stable during storage. Solutions prepared by extraction at both tested temperatures exhibited similar tolerance to 0.1 mol/L salt addition and freeze-thaw cycles. Moreover, the higher extraction temperature produced a FG solution with superior foaming and emulsification properties, and these properties were more stable with storage. Foams and emulsions produced from FG extracted at higher temperatures also had better stability. FG extracted at 98 °C displayed improved stability and consistent viscosity, foamability, and emulsification properties in comparison to solutions prepared at 70 °C. Therefore, the FG solution extracted at 98 °C had more stable properties and, potentially, higher commercial value. This result indicates that FG performance as a commercial food additive can influence food product quality.

## 1. Introduction

Flaxseed (*Linum usitatissimum* L.) is a member of the genus *Linum* in the family Linaceae [1]. It is an important oilseed crop used in food, feed, and industrial applications [2]. According to the latest annual flaxseed production records available in the statistical database of the Food and Agriculture Organization of the United Nations (FAOSTAT), Canada, China, Russia, the United States, Kazakhstan, India, and Ethiopia have produced the majority of flaxseed (2.41 million tonnes) over the last 20 years [3]. Flaxseed was introduced to North America for oil and fiber production; it is finding use for its health benefits as a nutraceutical with anti-atheroschlerosis, anti-cancer, and anti-inflammatory and anti-osteoporosis properties [4,5,6]. Commercial brown Canadian flaxseed has been reported to contain 41% fat, 20% protein, 28% total dietary fiber, 7.7% moisture, and 3.4% ash [7]. Flaxseed contains various nutrients such as alpha-linolenic acid and bioactive compounds. It has been classified as a functional food or bioactive food [8].

Flaxseed gum (FG) is a soluble dietary fiber [9] present in flaxseed hull (Figure 1). FG is composed of 80% neutral and acidic polysaccharide and 9% protein [10]. The neutral monosaccharides consist of xylose, glucose, arabinose, and galactose, while acidic monosaccharides consist of rhamnose, galactose, fucose, and galacturonic acid [10]. The protein in FG is predominantly conlinin [11]. The content of FG in seed varies between 10–15% of seed mass and its composition depends on cultivar, production practices, environment, and storage conditions [11,12]. Fractions containing FG are commercial hydrocolloid polysaccharides that are increasingly being used in foods as the gums impart useful properties to foods and solutions [13].

As a commercial food hydrocolloid, FG produces solutions with higher foam capacity and better foam stability than similar solutions prepared with gum Arabic, guar gum, and xanthan gum at the same concentration [14]. Its foam capacity and foam stability increase with increased concentration and reach a maximum with 1% (w/v) gum [15]. FG is traditionally well known for its thickening and gelation behavior, however, it also influences the dispersed system properties through their interfacial properties. Emulsions are unstable multiphase system but the addition of emulsifier can significantly increase emulsion system stability. For dissolved FG, the presence of high-content, high-molecular weight (MW) polysaccharides and attached protein increases viscosity of the continuous solution phase and inhibits the collision and migration of dispersed particles; thus, emulsions can be stabilized. Gum emulsion system stability has been referred to as non-absorbing depletion stability [16]. Compared to other emulsifiers such as carrageenan, FG emulsions have higher emulsion stability and a wide range of water-in-oil phase ratios [17]. FG emulsification properties are determined by average gum MW, dissolving temperature, emulsifying temperature, oil-to-water ratio, and storage conditions. Increased average MW and dissolving temperature will lead to increased emulsification properties while increased oil-to-water ratio and temperature for emulsification and storage will result in lower emulsion stability [18]. Negative charges on FG biopolymers impart both water-holding capacity and amphiphilic properties. As such, FG and FG-rich fractions are applied in the food industry as a food additive in meat, dairy, gel, beverage, and flour products where it acts as an agent that thickens, emulsifies, stabilizes foams, and forms gels [19].

Concentrated dissolved FG is prepared by hot water extraction then used in food production. Due to the complicated polymer structure of FG and the presence of bioactive compounds, extraction temperature plays an important role in gum appearance, physicochemical properties, and functional properties [9]. For example, lower-temperature extraction can avoid the denaturation of bioactive compounds and save energy costs but may give lower gum yield, while higher-temperature extraction can inactivate microorganisms and enzymes but may lead to the formation of undesirable compounds and protein denaturation. 

This research compared FG appearance, functional properties, and stability between two extraction temperatures. The one with higher commercial value and better gum quality was determined.

## 2. Materials and Methods

### 2.1. Materials and Chemicals

Raw samples used in this research were the cultivated Canadian flaxseed cultivar ‘Crop Development Centre’ (CDC) Bethune harvested in 2015 from a farm in Saskatchewan (Corning, SK, Canada). Seed was stored at room temperature (22 °C) until analyzed.

### 2.2. FG Sample Preparation

#### 2.2.1. Hot Water Extraction

Xing et al. (2015) [20] investigated effects of extraction temperature on the rheological properties of FG. They reported that the content of polysaccharide and protein in the gel increased with temperature and reached maximum at 70 °C. Their finding was similar to that of Cui et al. (1994) [21], who found, in a response surface analysis, that maximum gum extraction occurred at 70 °C. Viscosity of gum extracted at 70 °C and 80 °C showed no obvious difference [20]. Therefore, 70 °C was selected as one of the extraction temperatures. Since 70 °C was not high enough to kill all microorganisms and inactivate all enzymes present in gum, a higher temperature (98 °C) was selected to further investigate gum appearance and other properties. To be specific, one liter distilled water was heated to 70 °C, then 100 g CDC Bethune flaxseed was added. The mixture was maintained at 70 °C with stirring for 1 h, and FG was separated from the seed by filtering through cotton cheesecloth (Fischer Scientific, Edmonton, AB, Canada). Another liter of distilled water was heated to 70 °C, then the previously filtered flaxseed was added to the hot water and heated with stirring for an additional hour. The second FG portion was filtered through cotton cheesecloth and mixed with the first portion to obtain a water-extracted FG fraction 70 °C (w/v, 1/20). Using the same method, FG was prepared at 98 °C where water was heated to 98 °C [22].

#### 2.2.2. Sample Separation

Extracted FG samples were stored at 4 °C for eight days in total. At zero, one, two, four, and eight days, a small portion of each sample (400 mL) was collected and frozen at −28 °C until analyzed. For some analyses, FG was first precipitated by anhydrous ethanol with a ratio of 1:3 (FG extract:ethanol) followed by centrifugation (Allegra X-22R Centrifuge, Beckman Coulter, Mississauga, ON, Canada). Subsequently, the pellet was freeze-dried to obtain dried FG.

### 2.3. Physicochemical Properties

#### 2.3.1. FG Yield 

A separate extraction experiment was performed to determine the FG yield at two different temperatures. The extraction process was similar to that described in hot water extraction of FG, except the seed weight was 20 g and 400 mL of water was used. Extracted FG was precipitated by anhydrous ethanol and dried by freeze-drying for 15 h. The FG yield can be calculated using Equation (1).
FG yield% = (FG weight after freeze drying/seed weight) × 100(1)

#### 2.3.2. The pH Measurement

The pH of FG solutions was measured using a pH meter (ORION 4 STAR, Thermo Fisher Scientific, Waltham, MA, USA) calibrated by pH 4.0, 7.0, 10.0 buffer solutions. The pH values were obtained at zero, one, two, four, and eight days of 4 °C storage.

#### 2.3.3. Color Measurement

FG color coordinates (*L**, *a**, *b**) were determined using a ColorFlex EZ spectrophotometer (4510, HunterLab, Reston, VA, USA). Color parameter *L** indicates the lightness with range from 0 (brightness) to 100 (darkness); *a** refers to redness in positive and greenness in negative; *b** shows the level of yellowness in positive and blueness in negative value [23]. The color spectrophotometer was initially standardized at D65/10° setting using black glass and white tile (ISO 11475:2017). Then the sample was placed in a petri dish and covered by a black cap to eliminate stray light. Three measurements were taken with rotating cup 90° between each measurement.

#### 2.3.4. Optical Density Measurement

The optical density (OD) of FG solution was determined by measurements at 420 nm (OD_420_) using an ultraviolet–visible (UV-VIS) spectrophotometer (GENESYS 10S, Thermo Scientific, Madison, WI, USA). The spectrophotometer was adjusted to 420 nm. Distilled water was used as blank. The sample was transferred into a cuvette and OD_420_ was recorded. FG readily formed coacervates with protein and, as it has a natural amount of protein present, it was thought that FG extracts were coacervates. Light scattering observed in fresh FG was evidence of the formation of a coacervate.

#### 2.3.5. Viscosity Measurement

FG solution viscosity was determined by the shell cup method (shell cup #3.5 and #2, Norcross Shell Cup Viscometer, Norcross Corp., Newton, MA, USA). A shell cup was submerged in the FG for 30 s to allow the cup to come to the sample temperature. Then the cup was lifted vertically out of the fluid. The time to drain the cup was recorded from the time the cup broke the surface until the stream broke. Time required for the cup to empty was recorded. Viscosity of FG was obtained from appropriate conversion charts (See Note 2 and Note 3 in reference table) [24]. Sample viscosity was obtained at zero, one, two, four, and eight days of storage of 4 °C. A 50-mL portion of FG solution was mixed well with 10 mL 0.1 Molar (M) sodium chloride (NaCl) solution (≥99.5%, Sigma-Aldrich Canada Ltd., Oakville, ON, Canada). The mixture was kept static for 1 h and the viscosity was then measured using the shell cup. The results were compared to FG solution viscosity to determine the effects of salt addition on FG solution viscosity and changes during storage. FG solution was frozen at −28 °C for 24 h and thawed for viscosity measurement by the shell cup method. The results were compared to FG solution viscosity to analyze effects of freezing on FG viscosity.

#### 2.3.6. Foamability Analysis

FG solution was stirred by a high-speed agitator (1500 rpm, KHM512IC 5-Speed Ultra Power Hand Mixer, Whirlpool Corp., Benton Charter Township, MI, USA) until a persistent foam formed (5 min). FG-foam volume was measured immediately after agitation and after keeping static for 24 h [15,25]. Foamability was measured by calculation of %foaming and %foam stability using Equations (2) and (3), respectively.
(2)$$\%Foaming=\frac{Foam\;Volume}{Gum\;Volume}\times 100$$
(3)$$\%Foam\;Stability=\frac{Foam\;Volume\;after\;24\;Hours}{Foam\;Volume\;right\;after\;Agitation}\times 100$$

#### 2.3.7. Emulsification Property Analysis

FG can be used in food emulsion systems to prevent aggregation of dispersed oil droplets in the aqueous phase. FG is not a true emulsifier since it stabilizes emulsions primarily by thickening and increasing the viscosity of the aqueous layer and minimizes the tendency of oil droplets to migrate [13]. The emulsification capacity and emulsion stability of FG-based emulsions depend on several factors such as emulsifying temperature, time, and gum quality [19]. In this research, soybean oil (Crisco, The J.M. Smucker Company, Orrville, OH, USA) was dispersed in gum solution with oil-to-gum ratio of 1:10 for 1 min at room temperature. The emulsion was transferred into a 100-mL measuring cylinder and kept static for 2 h to determine its emulsification capacity. Emulsion stability was determined by measuring emulsion layers after 2 days. Under these conditions, the thicker the emulsion layer, the better the emulsification capacity was [26]. Emulsification capacity and emulsion stability can be calculated using Equations (4) and (5), respectively.
(4)$$Emulsion\;Ratio\;\left(\%\right)=\frac{Total\;Solution\;Volume-Aqueous\;Layer\;Volume}{Total\;Solution\;Volume}\times 100$$
(5)$$Emulsion\;Stability\;\left(\%\right)=\frac{Emulsion\;Ratio\;after\;Keeping\;Static\;for\;2\;Days}{Emulsion\;Ratio\;after\;Keeping\;Static\;for\;4\;Hours}\times 100$$

### 2.4. Statistical Analysis

All statistical analyses were performed using the Statistical Package for the Social Sciences (SPSS) version 25.0 (IBM Corp., Armonk, NY, USA). Mean comparisons were made by analysis of variance (ANOVA) and Tukey’s post hoc test or comparisons of means of two groups, using an independent sample *t*-tests. Data are presented as mean ± standard deviation (SD) (*n* = 3), and *p* < 0.05 was considered statistically significant.

## 3. Results and Discussion

The production costs and quality stability of FG are very important in determining its commercial value as a food additive. Several main characteristics of FG extracted at two different temperatures were analyzed and compared in this research to determine their impact on overall quality during storage. Extraction of FG from whole seed with hot water is a practical method for removing FG from seed.

### 3.1. FG Yield and pH

The yield of FG from water extraction was determined at 70 °C and 98 °C to be 10.97 ± 1.07 g/100 g flaxseed and 12.73 ± 1.12 g/100 g flaxseed, respectively. The FG yield from extraction at 70 °C was lower than at 98 °C. In research by others, gum yield also increased with increased extraction temperature [22]. Liu et al. (2016) [11] used extraction methods similar to those described here at 60 °C and found that the gum yield from CDC Bethune flaxseed (the cultivar used in this study) was 9.3%. This result is similar to the findings of the current study where higher temperatures produced greater recoveries.

The pH of FG solutions has a significant effect on physicochemical properties including flow behavior and viscosity. Under acidic conditions, FG gel strength decreases as pH value decreases; under alkaline conditions, FG gel strength decreases as pH value increases [27,28]. It is important to analyze changes in FG solution properties with pH changes during storage as such variation might be undesirable. FG has optimal behavior at a pH values from 6.0–8.0 [29]. The pH value of FG solutions recovered by extraction at 70 °C and 98 °C were recorded during eight days of storage at 4 °C (Figure 2). During storage, FG solutions had pH values between 6.1–6.3 with less than 0.1 unit change over time. Thus, FG extracted at both temperatures would likely have suitable pH stability for use as a food ingredient. The pH of FG solution extracted at 98 °C was slightly higher than that of FG solution extracted at 70 °C. This observation can likely be explained by the denaturation of proteins and polysaccharides due to high-temperature extraction [22]. In this research, gum pH values were relatively stable over storage at a pH that would be considered optimal.

### 3.2. Color Analysis

As FG would likely be used as either a food or beverage additive or component, its color is an important factor as this affects the appearance of products, especially drinks. While the color of FG powders has been reported, the color of FG solution has not been reported in literature [30]. Two FG solutions (70 °C and 98 °C) were analyzed for their initial color and changes in color during the storage period of eight days based on three color parameters (*L**, *a***,* and *b**). Table 1 indicates that *L* value increased significantly (*p* < 0.05) from 11.5 to 14.9 at 70 °C and from 16.7 to 18.4 at 98 °C during storage. FG prepared at 98 °C had a higher initial *L** (16.69 ± 0.11) or darker color than FG (11.53 ± 0.15) prepared at 70 °C. As FG contains both protein and carbohydrate, it is possible that the darkening was related to a Maillard reaction between these constituents. During storage, the increased *L** value for these samples indicated that both were darkening with increased storage. Parameter *a** did not change significantly with storage. FG prepared at 70 °C was less green than FG prepared at 98 °C. The *b** values of FG prepared at 70 °C were in the range of −1.77 ± 0.12 to −2.41 ± 0.03 while that of 98 °C FG were in range of 2.21–3.58. These results indicated light blueness in the 70 °C FG while light yellowness was more prevalent in the FG prepared at 98 °C. Blueness in 70 °C FG increased with increased storage time. In contrast, yellowness in 98 °C FG became lighter during storage.

### 3.3. Light Scattering and Absorption

FG solutions attenuate light by both scattering and adsorption. The OD_420_, a measure of the sum of both scattering and absorption, was determined by measuring the attenuation of light at 420 nm passing through FG solutions (Figure 3). The OD_420_ of FG extracted at 70 °C started at 0.571 and increased to 0.637 over eight days of storage at 4 °C, while the apparent OD_420_ of FG prepared at 98 °C increased from 0.711 to 0.854. These results indicated that FG solutions prepared at 70 °C scattered less incident light (420 nm) than solutions prepared at 98 °C. As flaxseed has protein that could form a natural coacervate, it is possible that the lower OD_420_ might have been due to a lower content of protein in the extract made at the lower temperature and lower amount of coacervate formed. Moreover, the increased apparent OD_420_ value of 70 °C FG was also lower than that of 98 °C FG during storage. Combining the result of color analysis and attenuation of transmittance, it was apparent that FG prepared by extraction at 70 °C produced a more transparent solution with superior color when compared to solutions extracted at 98 °C.

### 3.4. Viscosity Analysis

Viscosity is an important measure of the utility of food thickeners and emulsifiers like FG. Viscosity can be influenced by many factors such as extraction temperature and storage conditions. In this research, FG solutions from the samples extracted at two temperatures were analyzed to determine viscosity changes during storage. In addition, the effects of salt and freezing on these solutions was determined.

#### 3.4.1. Viscosity During Storage

Viscosity changes in FG solutions were determined over eight days of storage at 4 °C and the viscosity of both samples decreased during the storage period (Figure 4). FG solutions prepared from gum extracted at 70 °C had higher viscosity than solutions prepared at 98 °C (96.7 mPa·s vs. 78.8, respectively) in the first day after extraction but lower viscosity than 98 °C (70.1 mPa·s vs 71.9 mPa·s) after storage at 4 °C for eight days. These differences are explained by the composition of the FG extracts produced at the two temperatures. When extracting FG at 70 °C, most of the high MW molecules were dissolved into water, which gave FG high initial viscosity. However, the temperature was not high enough to inactivate all enzymes and microorganisms. It is likely that higher MW polymers were metabolized to smaller molecules during storage and viscosity was decreased [31]. Conversely, the higher temperature extraction (98 °C) likely hydrolyzed bonds in a portion of the high MW polymers during extraction. This produced an FG extract with relatively lower initial viscosity. However during the storage period, the inactivated enzyme and bacteria could not further degrade the gum. This led to the observation of more stable viscosity.

#### 3.4.2. Effect of Salt (NaCl) on Viscosity

A significant portion of FG is an anionic polysaccharide. It bears negative charges due to ionized carboxyl group. The electrostatic repulsion between molecules with the same charge induces molecular chains to extend and interpenetrate each other in solution to increase viscosity [28,32]. The addition of salt can screen the charge interactions between protein and carbohydrate biopolymers and, thereby, decrease viscosity [9]. As salt is widely used in food production for flavor improvement, knowledge of the effect of salt on gum viscosity is important in predictions of gum behavior in food products. Salt addition affected both FG extracts over time (Figure 4). Addition of NaCl led to decreased (11 mPa·s) FG solution viscosity. Salt addition to FG solutions resulted in compression of the double electric layer of molecules, which lowered the intermolecular and intramolecular repulsion between ionic macromolecules like the negatively charged gums. The decrease in interaction between the molecules decreased viscosity [28]. The effect of salt addition on FG solution viscosity was not influenced by either extraction temperature or storage duration.

#### 3.4.3. Freeze-Thaw Effects on FG Viscosity

Freezing is widely applied in food production and storage and ice crystals produced during freezing can induce structure or property changes in food ingredients. For an ingredient to be used in frozen products it is important to determine ingredient behavior and appearance by comparing before and after a freeze-thaw cycle [26]. In this study, the viscosity was determined on FG solution samples stored for different times before freezing and after thawing. For FG solutions prepared by extraction at both 70 °C and 98 °C, freezing followed by thawing had no effect on FG viscosity (Figure 5). Some separation of solution was observed during thawing of FG solutions but the solutions were made homogenous by gently shaking before testing the viscosity. Therefore, FG solutions were not altered by freezing followed by thawing.

### 3.5. Foamability Analysis

Foam is a two-phase system consisting of water phase and air phase. Foaming agents have been widely applied in bakery products such as sponge cake and mousse to produce a fluffy texture [33]. FG hydrocolloid solutions have the property of foamability that is a measure of the ability of the polymer to stabilize foam and to limit air escape and foam collapse [32]. Foamability plays a critical role in FG application as a foaming agent in cooking. FG foam capacity and stability were determined by observations of foam volume over time. Foam capacity of the FG solution prepared at 70 °C increased with storage time from 127.2% to 133%, while the foam capacity of FG solution prepared at 98 °C was nearly constant during storage (Figure 6A). The FG solution prepared at 98 °C showed higher foam capacity than the solution prepared at 70 °C during storage. The difference in foam capacity decreased with increased storage duration and after 8 days of storage their foam capacities were similar. The smaller foam capacity of 70 °C FG might have been due to its high initial viscosity and high MW polysaccharide molecules which had lower capacity to stabilize interfaces in the foam and resist collapse. With increased storage time, enzymes and microorganisms present might lower the average MW and increase foam capacity over time. The higher extraction temperature of 98 °C might decrease the average polysaccharide molecule size and lower the average MW [15]. The smaller molecules in the FG solution prepared at provided 98 °C might have caused the higher foam capacity that was observed.

Foam stability of FG solutions prepared at 70 °C increased from 79.7% to 87.4% with increased FG solution storage while the foam stability of solutions prepared at 98 °C was constant (87.5%, Figure 6B). In summary, throughout the eight-day storage, the solution FG extracted at 70 °C exhibited increased foam capacity and foam stability while the FG solution extracted at 98 °C exhibited constant foam capacity and stability. Subjectively, the FG solution prepared at 98 °C produced a foam with better appearance than the foam from solutions prepared at the lower temperature.

### 3.6. Emulsification Capacity and Emulsion Stability

Emulsification can be induced when high MW molecules form a protective film at oil-water interfaces and, thereby, lower the interfacial tension between oil and water [15]. In this research, sample emulsifying capacity was determined by measuring emulsion layer ratio and change of this ratio over time.

Immediately after extraction the emulsion layer of the solution prepared at 70 °C (45.3%) was greater than observed for the solution prepared at 98 °C (43%, Figure 7A). However, with increased storage the emulsification capacity of the solution prepared at 70 °C decreased continuously while the solution prepared at 98 °C maintained a constant emulsification capacity. Emulsification capacities of the solution prepared at 70 °C exhibited a relatively higher decrease after 4 days and 8 days of storage. The higher emulsification capacity of the solution prepared at 70 °C is likely due to the higher average MW of polysaccharides in this sample that imparted a higher initial solution viscosity. The higher viscosity limited oil droplet migration and the observed higher emulsification capacity [34]. In the presence of enzymes and bacteria, the molecular weight of solution hydrocolloids will decrease and the viscosity of the FG solution decreases. Oil droplet migration rates in solution would increase, as observed in the FG solution. Emulsification capacity decreased, as observed in the solution prepared at 70 °C. For the FG solution prepared at 98 °C, the average gum MW was likely smaller. The extracted protein and other impurities that were dissolved at the higher temperature might have limited oil migration to produce a solution with relatively high emulsification capacity [26]. Moreover, without the side effects of enzymes and microorganisms, the emulsification capacity was maintained during the storage period.

Emulsion stability measurements were performed on FG solutions on the 0, 1st, 2nd, 4th, and 8th day of 4 °C storage (Figure 7B). Emulsion stability of FG solutions prepared at 98 °C were initially 99.5% immediately after extraction and fell to 98.7% after eight days of storage. FG solutions prepared at 70 °C had an initial stability of 96.9% and final stability of 91.4%. This result indicated that emulsifying stability of FG solutions prepared at 98 °C was higher and more constant during the storage period than observed for solutions prepared at 70 °C. Combining the results of emulsifying capacity, FG solutions extracted at 98 °C showed better overall emulsibility than solutions prepared at 70 °C.

## 4. Conclusions

This work evaluated the functionality of FG obtained after hot water extraction at either 70 or 98 °C. The higher temperature extraction enabled a higher yield of FG with improved emulsification properties and enhanced emulsion stability when compared with FG prepared at the lower extraction temperature. Unfortunately, the product obtained at the higher temperature was darker than that recovered at the lower temperature. These results indicated the need for further studies to improve extraction conditions that can improve FG yield and quality, as well as investigations of the potential for incorporating FG in food products.

## Figures and Tables

**Figure 1 foods-09-00681-f001:**
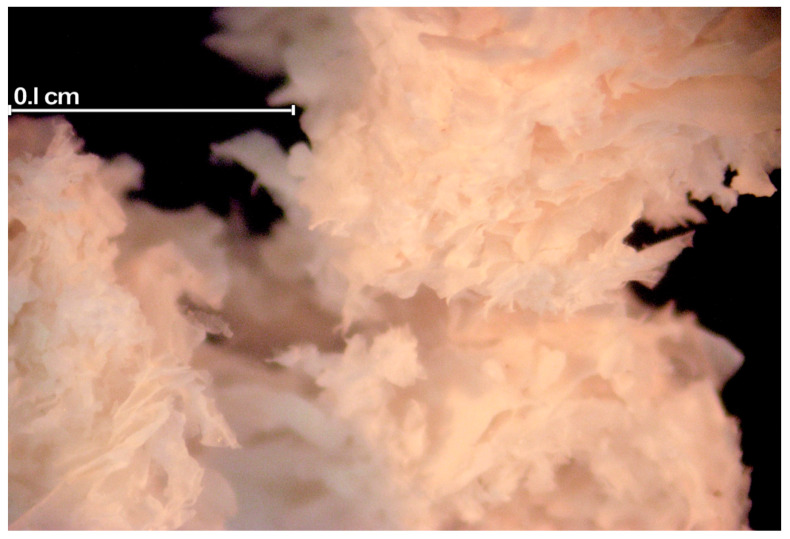
Flaxseed gum [FG, *Linum usitatissimum* L., var. Crop Development Centre (CDC) Bethune]. Images were obtained (1000 magnification) with a Canon Eos 300D digital camera mounted on a Zeiss Stemi SV 11 light microscope (Carl Zeiss AG, Oberkochen, Germany). The images were subsequently processed in Photoshop 7.

**Figure 2 foods-09-00681-f002:**
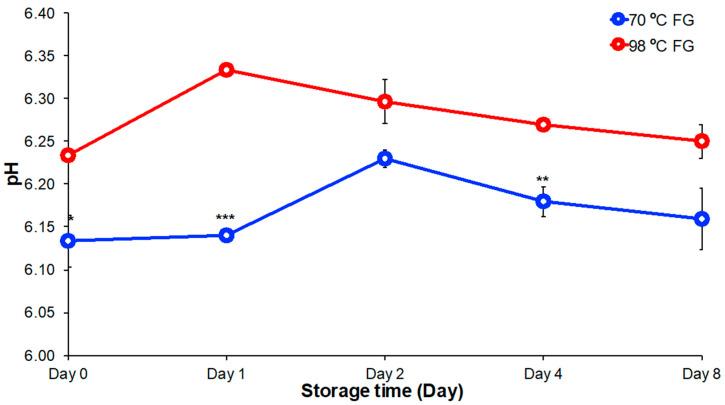
The pH change of FG extracted at 70 °C and 98 °C over for eight days of storage at 4 °C. Significant difference indicates at * *p* < 0.05, ** *p* < 0.01, and *** *p* < 0.001 by two independent samples.

**Figure 3 foods-09-00681-f003:**
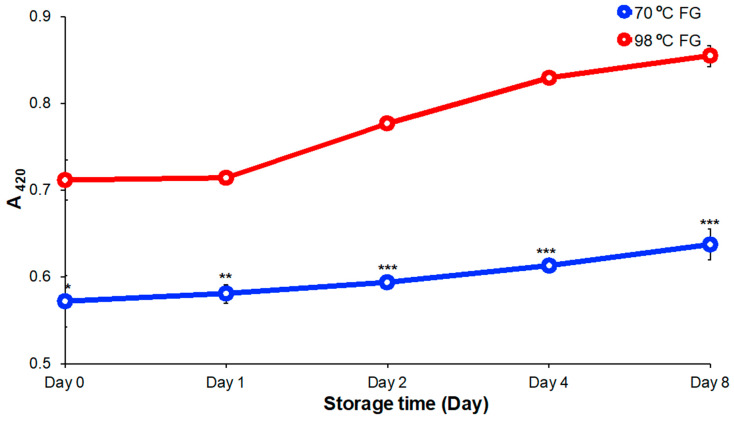
Optical density (OD) at 420 nm (OD_420_) change of FG extracted at 70 °C and 98 °C for eight days of storage at 4 °C. Significant difference indicated at * *p* < 0.05, ** *p* < 0.01, and *** *p* < 0.001 by two independent samples.

**Figure 4 foods-09-00681-f004:**
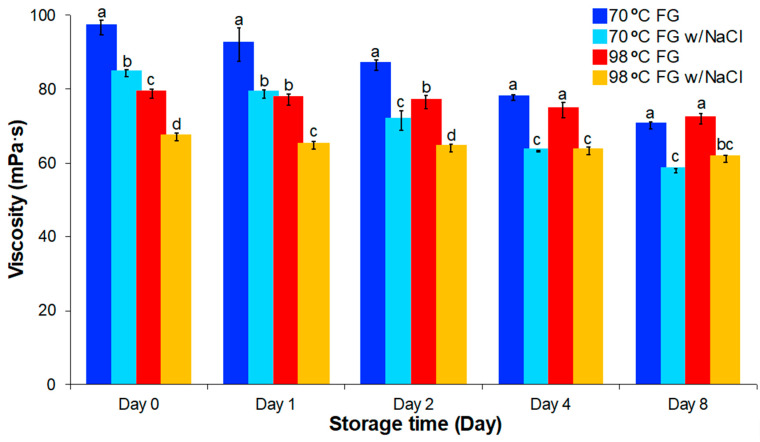
Viscosity change of FG with salt addition stored at 4 °C for different time (zero, one, two, four, and eight days) compared to FG solutions extracted at 70 °C and 98 °C. Means within the same day without a common letter (a–d) were significantly different according to Tukey’s post hoc test (*p* < 0.05).

**Figure 5 foods-09-00681-f005:**
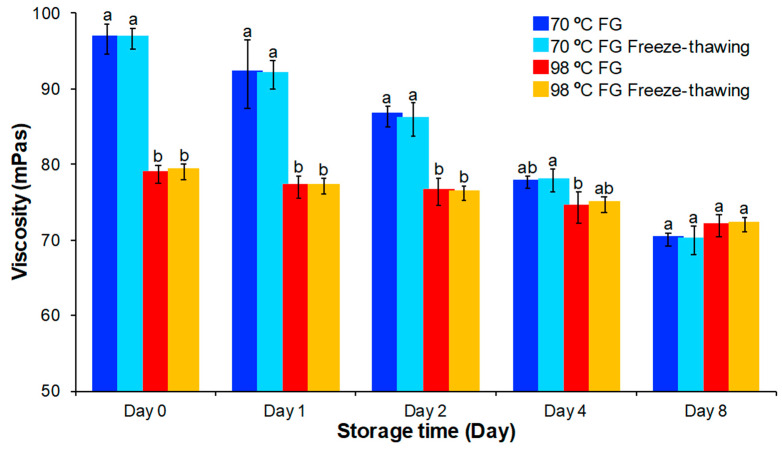
Effect of freeze-thawing on viscosity of FG compared to FG solutions extracted at 70 °C and 98 °C over eight days. Means within the same property without a common letter (a–d) were significantly different according to Tukey’s post hoc test (*p* < 0.05).

**Figure 6 foods-09-00681-f006:**
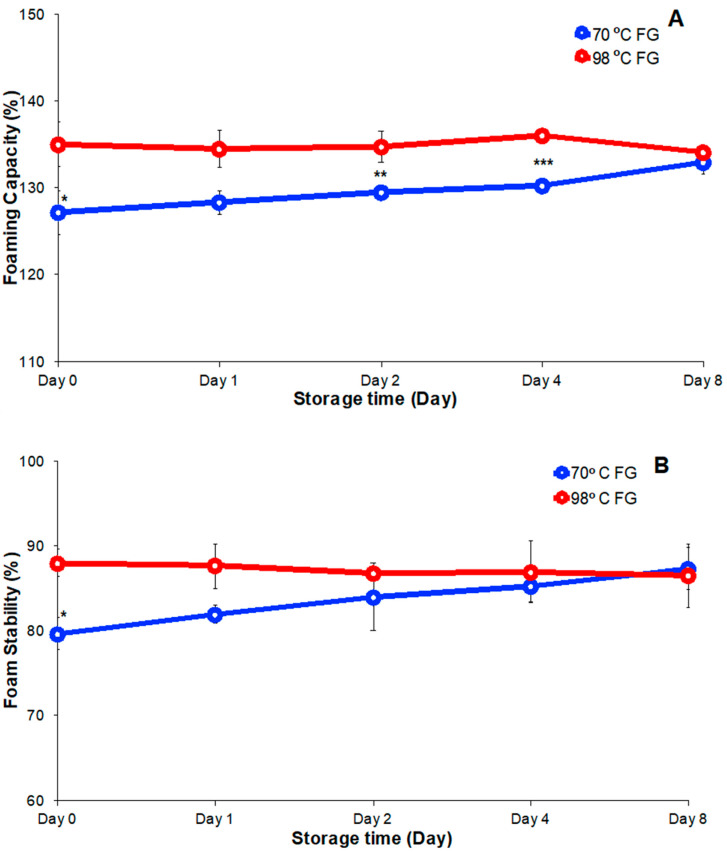
(**A**) Foam capacity and (**B**) stability change of FG over eight days of storage at 4 °C. Significant difference indicated at * *p* < 0.05, ** *p* < 0.01, and *** *p* < 0.001 by two independent samples.

**Figure 7 foods-09-00681-f007:**
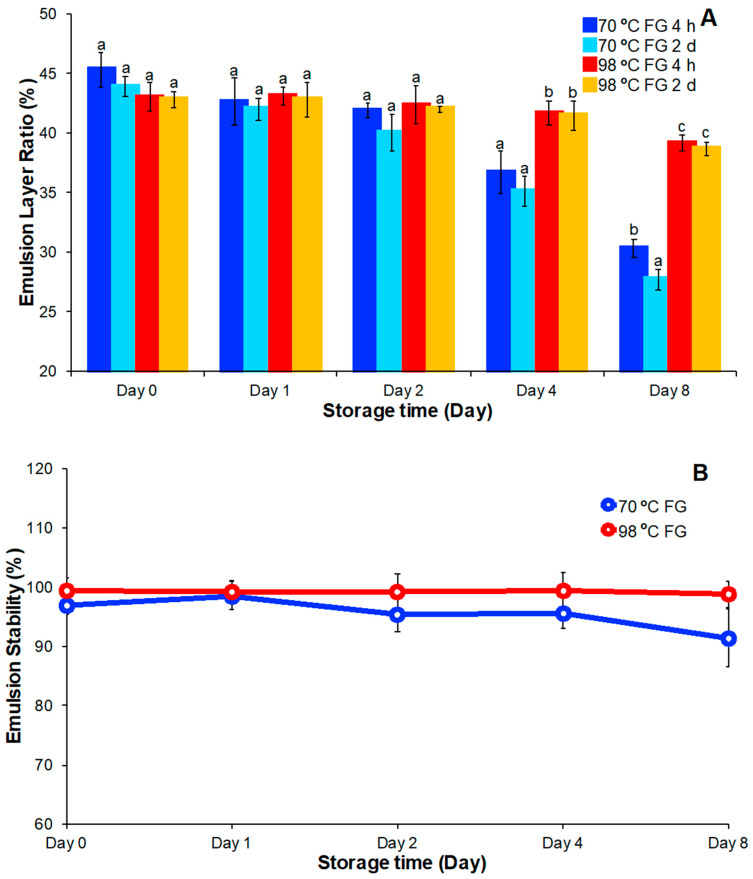
(**A**) Emulsion capacity and (**B**) stability change of FG over eight days of storage at 4 °C. Means within the same property without a common letter (a–c) were significantly different according to Tukey’s post hoc test (*p* < 0.05).

**Table 1 foods-09-00681-t001:** Color parameters of FG at 70 °C and 98 °C during storage.

Storage Time	*L**	*a**	*b**
**70 °C FG**			
**Day 0**	**11.53 ± 0.15 ^a^**	**−1.08 ± 0.04 ^d^**	**−1.77 ± 0.12 ^b^**
**Day 1**	**11.72 ± 0.15 ^a^**	**−1.21 ± 0.04 ^c^**	**−1.97 ± 0.06 ^b^**
**Day 2**	**12.47 ± 0.03 ^b^**	**−1.22 ± 0.23 ^c^**	**−2.26 ± 0.03 ^a^**
**Day 4**	**12.72 ± 0.03 ^b^**	**−1.15 ± 0.01 ^c,d^**	**−2.33 ± 0.06 ^a^**
**Day 8**	**14.92 ± 0.13 ^c^**	**−1.17 ± 0.02 ^c^**	**−2.41 ± 0.03 ^a^**
**98 °C FG**			
**Day 0**	**16.69 ± 0.11 ^d^**	**−1.61 ± 0.01 ^b^**	**3.58 ± 0.16 ^g^**
**Day 1**	**16.89 ± 0.07 ^d^**	**−1.73 ± 0.01 ^a^**	**3.21 ± 0.09 ^f^**
**Day 2**	**17.25 ± 0.06 ^e^**	**−1.74 ± 0.00 ^a^**	**2.82 ± 0.05 ^e^**
**Day 4**	**17.92 ± 0.04 ^f^**	**−1.72 ± 0.04 ^a^**	**2.54 ± 0.06 ^d^**
**Day 8**	**18.41 ± 0.08 ^g^**	**−1.67 ± 0.04 ^a,b^**	**2.27 ± 0.04 ^c^**

^a–g^ Values followed by different letter within a column are significantly different (*p* < 0.05) according to Tukey’s test. *L**, brightness/darkness; *a**, (+) redness/(–) greenness; and *b**, (+) yellowness/(–) blueness; FG, flaxseed gum.

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
