# Peer review of "Flaxseed Gum Solution Functional Properties"

_foods, 2020, doi:10.3390/foods9050681_

Round 1
Reviewer 1 Report
Dear authors,
The efforts done to evaluate the functionality of a product derived from flaxseed resulted in an interesting work, of interest to the scientific community. Nevertheless the manuscript present some weaknesses that I believe the authors can overcome to improve the quality of the manuscript
The manuscript must be updated. Of the total of 27 references cited, 20 old are old. Revise and rewrite the contextualization (the reasons that motivated your work) in the introduction section and compare your work with literature (that evaluated funcionality flaxeed gums obtained by other methods or the obtaining of natural gums derived from another sources by the use of hot water extraction) using preferably recent references (2015-2020).
Another observations
topic 2.3.5 - include specifications of equipment, chemicals etc (brand, model, city, country)
topic 2.3.6 - include specifications of equipment (brand, model, city, country) and the homogenization (type of rotor, rotational speed, time of homogenization)
topic 2.3.4 explain the method and calculation procedure to meaure transparency. Insert city and country of spectrophotometer
topic 2.3.5 This method is not clear. Explain the usability of the method and the procedure. What was the shear rate applied to your sample? What was the running time for the samples? The answer for this analysis is only the viscosity? What about the shear stress, shear rate? Did you evaluated the dependence of viscosity versus the shear time (thixotropy/rheopexy)? This method could provide you more precise answers for the viscosity decreasing presented in your samples during storage.
topic 2.3.7 insert city an country of soybean oil obtained
3.1 section
179-180 - the authors should explain the undesirable consequences in the FG at unoptimal pH conditions. It was not clear
Topic 3.2
The increasing of darkness and blueness is correlated to what? Compare your data with recent literature.
210-212 - The extinction coefficient (A420) must be firsty explained in the M&M section. For what does this parameter serve? It is not clear. Revise and rewrite the whole discussion.
Lower values of A420 mean what? Degradation? Compare your data with recent literature and insert a critical analysis of your data.
topic 3.4.1 Which other emulsification properties are you evaluating besides the viscosity? Revise the title of this section.
topic 3.4.2 For What does the electrostatic repulsion consist of? It was not clear. Explain the mechanism: why the addition of NaCl affect the interaction of gums?
topic 3.42 line 263-264 - Show the data; prove you results
line 268 - consisted; not ´´consists´´
topic 3.5 Insert the importance on the ability to form foams in food products
Additional comments
The authors should emphasize that hot water extraction is a low cost alternative to produce gums and the effect of this process on the functional properties of a new product (FG) was evaluated on the possibility to valorize flaxseed.
The authors should insert references on the obtaining of flaxseed gums by other methods (if any).
Author Response
Manuscript ID foods-769047 entitled “Flaxseed (Linum usitatissimum L.) Gum Solution Functional Properties”
Thank you for your patience. We have revised our manuscript (Manuscript ID: foods-769047) according to reviewers’ comments/suggestions and would like to thank all reviewers for their critical feedback in making this manuscript more polished. We have listed reviewers’ comments and answered them in sequence. In addition to the changes made to the manuscript as recommended by the reviewers, we have also tried to improve the manuscript’s structure. We appreciate the reviewers’ thoughtful comments and critiques and hope this response addresses the overall quality of this manuscript for publication.
Responses to Reviewer 1 Comments and Suggestions:
The efforts done to evaluate the functionality of a product derived from flaxseed resulted in an interesting work, of interest to the scientific community. Nevertheless, the manuscript presents some weaknesses that I believe the authors can overcome to improve the quality of the manuscript.
The manuscript must be updated. Of the total of 27 references cited, 20 old are old. Revise and rewrite the contextualization (the reasons that motivated your work) in the introduction section and compare your work with literature (that evaluated functionality flaxseed gums obtained by other methods or the obtaining of natural gums derived from another sources by the use of hot water extraction) using preferably recent references (2015-2020).
Response: We have revised as described and added six of the recent references on page 13.
Comment: topic 2.3.5 - include specifications of equipment, chemicals etc. (brand, model, city, country).
Response: Inserted: shell cup #3.5 and #2, Norcross Shell Cup Viscometer, Norcross Corp., Newton, MA, USA… sodium chloride (NaCl) solution (³99.5%, Sigma-Aldrich Canada Ltd., Oakville, ON, Canada)
Comment: topic 2.3.6 - include specifications of equipment (brand, model, city, country) and the homogenization (type of rotor, rotational speed, time of homogenization).
Response: Inserted: FG solution was stirred by a high-speed agitator (1500 rpm, KHM512IC 5-Speed Ultra Power Hand Mixer, Whirlpool Corp., Benton Charter Township, MI, USA)
Comment: topic 2.3.4 explain the method and calculation procedure to measure transparency. Insert city and country of spectrophotometer.
Response: Revised paragraphs as followings: The optical density of FG solution was determined by measurements at 420 nm (OD420) using an ultraviolet-visible (UV-Vis) spectrophotometer (GENESYS 10S, Thermo Scientific, Madison, WI, USA). The spectrophotometer was adjusted to 420 nm; distilled water was used as blank; the sample was transferred into a cuvette and OD420 was recorded. FG readily forms coacervates with protein and as it has a natural amount of coacervate present it is thought that FG extracts are coacervates. Light scattering observed in freshly FG is evidence of the formation of a coacervate.
Comment: topic 2.3.5 This method is not clear. Explain the usability of the method and the procedure. What was the shear rate applied to your sample? What was the running time for the samples? The answer for this analysis is only the viscosity? What about the shear stress, shear rate? Did you evaluate the dependence of viscosity versus the shear time (thixotropy/rheopexy)? This method could provide you more precise answers for the viscosity decreasing presented in your samples during storage.
Response: The reviewer’s comment is well taken but the research was not designed to answer the question posed by the reviewer. A future study could investigate rheology changes as described by the reviewer. We did not choose to present such data here.
Comment: topic 2.3.7 insert city a country of soybean oil obtained.
Response: soybean oil (Crisco, The J.M. Smucker Company, Orrville, OH, USA)
Comment: 3.1 section, 179-180 - the authors should explain the undesirable consequences in the FG at unoptimal pH conditions. It was not clear.
Response: We inserted a sentence: Under acidic conditions, FG gel strength decreases as pH value decreases; under alkaline conditions, FG gel strength decreases as pH value increases [26,27].
Comment: topic 3.2 - The increasing of darkness and blueness is correlated to what? Compare your data with recent literature.
Response: Discussed possible browning reaction. We cited Ref 29. “Moczkowska, M.; Karp, S.; Niu, Y.; Kurek, M.A. Enzymatic, enzymatic-ultrasonic and alkaline extraction of soluble dietary fibre from flaxseed–A physicochemical approach. Food hydrocolloids 2019, 90, 105–112.”
Comment: 210-212 - The extinction coefficient (A420) must be first explained in the M&M section. For what does this parameter serve? It is not clear. Revise and rewrite the whole discussion.
Response: The discussion was revised substantially as light absorption is not a major contribution to optical density. Therefore, this language was used consistently through the document and absorbance was avoided as a term. This has increased the clarity of the manuscript.
Comment: Lower values of A420 mean what? Degradation? Compare your data with recent literature and insert a critical analysis of your data.
Response: The OD is a result of the formation of a natural coacervate of flaxseed protein with flaxseed gum. The Coacervate particles scatter light and block to passage of light. We inserted a sentence as followings:
As flaxseed has protein that could form a natural coacervate it is possible that the lower OD420 might have been due to a lower content of protein in the extract made at the lower temperature and lower amount of coacervate formed.
Comment: topic 3.4.1 Which other emulsification properties are you evaluating besides the viscosity? Revise the title of this section.
Response: Changed the title: Viscosity During Storage
Comment: topic 3.4.2 For What does the electrostatic repulsion consist of? It was not clear. Explain the mechanism: why the addition of NaCl affect the interaction of gums?
Response: Revised all paragraphs and inserted Refs. 8 and 31.
The addition of salt can screen the charge interactions between protein and carbohydrate biopolymers and thereby decrease viscosity.
- Liu, J.; Shim, Y.Y.; Tse, T.J.; Wang, Y.; Reaney, M.J.T. Flaxseed gum a versatile natural hydrocolloid for food and non-food applications. Trends Food Sci. Technol. 2018, 75, 146–157.
- Huang, L. Food Thickener, 2nd ed.; Huang, L., Ed.; China Light Industry Press: Beijing, China: 2009.
Comment: topic 3.4.2 line 263-264 - Show the data; prove you results.
Response: Inserted
Comment: line 268 - consisted; not ´´consists´´
Response: fixed
Comment: topic 3.5 Insert the importance on the ability to form foams in food products
Response: Inserted and revised:
Foaming agents have been widely applied in bakery products such as sponge cake and mousse to supply the fluffy texture [31]. … Idea foamability plays a critical role in FG application as a foaming agent in cooking.
Additional Comments: The authors should emphasize that hot water extraction is a low-cost alternative to produce gums and the effect of this process on the functional properties of a new product (FG) was evaluated on the possibility to valorize flaxseed.
Response: Revised paragraphs as followings: “Extraction of FG from whole seed with hot water is a practical method for removing FG from seed.”
Additional Comments: The authors should insert references on the obtaining of flaxseed gums by other methods (if any).
Response: Several such references are now included describing extraction conditions used by others.

Reviewer 2 Report
In this study, the effect of extraction temperature (70 and 98°C) on the functional properties and storage stability of flaxseed mucilage was investigated. The research was interesting and the writing and organization of the manuscript were acceptable. However, there remained some problems about the format as well as the samples. Consequently, the manuscript is recommended for a minor revision and the issues need to be addressed before publication.
- In the Introduction,
-Line 53, a comma is needed after hydrocolloid and before FG.
Line 53-55, Is it really documented that FG solutions has a better foam capacity and foam stability in comparison to Arabic gum at same concentration??!If there is any, please mention the reference?
-There are some recent researches on the characterization and application of flaxseed gum, so, I would suggest that the authors add these works to their introduction including,
https://doi.org/10.1016/j.ijbiomac.2020.02.149
https://doi.org/10.1016/j.colsurfa.2018.12.004
https://doi.org/10.1016/j.foodres.2019.108779
https://doi.org/10.1016/j.colsurfb.2019.110489
- In Material and methods,
-Why only two temperatures were selected for the evaluation? What about the temperatures below 70 or between 70 and 98°C? especially 80 °C as it is the most common temperature in the literature for the extraction of flaxseed gum.
- Why the effect of different pH values during gum extraction was not evaluated?
- Since the presence of protein in the structure of the flaxseed gum is responsible for its interfacial properties such as emulsifying ability and foaming ability, authors should have considered the effect of extraction temperature on the amount of the protein in the extracted gum. The content of the protein in the extracted gums at different temperatures should be measured and its influence on the gum functional properties should be discussed?
- In Results and Discussion,
-Line 217, a comma is needed after transmittance and before it.
- Line 227, a comma is needed after In addition and before the.
- Line 229, the title " Emulsification Properties" doesn’t seem to be for the mentioned discussion since the discussion is on the viscosity rather that emulsification properties.
- line 293-319, Since the FG extracted at higher temperature possessed better surface active properties, it is suggested that the effect of protein level in the two extracted gums will be discussed.
- In general, the discussion did not cover adequately the results of other researchers and it is essential to include and compare the findings of other studies, especially, new articles.
- As a conclusion despite the interesting work, I recommend revision based on the mentioned above comments for publication in Foods. Manuscript should be rewritten to make it concise and jargon free.
Author Response
Responses to Reviewer 2 Comments and Suggestions:
In this study, the effect of extraction temperature (70 and 98°C) on the functional properties and storage stability of flaxseed mucilage was investigated. The research was interesting, and the writing and organization of the manuscript were acceptable. However, there remained some problems about the format as well as the samples. Consequently, the manuscript is recommended for a minor revision and the issues need to be addressed before publication.
- In the Introduction,
Comment: Line 53, a comma is needed after hydrocolloid and before FG.
Response: fixed.
Comment: Line 53-55, Is it really documented that FG solutions has a better foam capacity and foam stability in comparison to Arabic gum at same concentration??!If there is any, please mention the reference?
Response: Inserted Ref 13.
- Chen, H.; Xu, S.; Wang, Z. Film and foam properties of flaxseed gum. Food and Fermentation Industries. 2006, 32(6), 34-36.
Comment: There are some recent researches on the characterization and application of flaxseed gum, so, I would suggest that the authors add these works to their introduction including, https://doi.org/10.1016/j.ijbiomac.2020.02.149
https://doi.org/10.1016/j.colsurfa.2018.12.004
https://doi.org/10.1016/j.foodres.2019.108779
https://doi.org/10.1016/j.colsurfb.2019.110489
Response: We inserted 2 Refs as followings:
- Liu, J.; Shim, Y.Y.; Shen, J.; Wang, Y.; Ghosh, S.; Reaney, M.J.T. Variation of composition and functional properties of gum from six Canadian flaxseed (Linum usitatissimum L.) cultivars. Int. J. Food Sci. Tech. 2016, 51(10), 2313–2326.
- Chen, H.; Xu, S.; Wang, Z. Film and foam properties of flaxseed gum. Food Ferment. Ind. 2006, 32(6), 34–36.
- In Material and methods,
Comment: Why only two temperatures were selected for the evaluation? What about the temperatures below 70 or between 70 and 98°C? especially 80 °C as it is the most common temperature in the literature for the extraction of flaxseed gum.
Response: Inserted/ revised as below:
Xing et al. (2015) [19] investigated the effects of extraction temperature on the rheological properties of FG. They reported that the content of polysaccharide and protein in the gel increased with the temperature increasing and reached maximum at 70 oC. Their finding was similar to that of Cui et al. (1994) [20] who found in a response surface analysis that maximum gum extraction occurred at 70 oC. The viscosity of gum extracted at 70 oC and 80 oC showed no obvious difference [19]. Therefore, 70 oC was selected as one of the extraction temperatures. Since 70 oC was not high enough to deactivate microorganism and enzyme presented in gum, higher temperature (98 oC) was selected to further investigate gum appearance and other properties.
Comment: Why the effect of different pH values during gum extraction was not evaluated?
Response: Gum was extracted with the same condition with only variations in temperature tested. There are many other manuscripts that describe the effect of pH on gum extraction. We did not feel the need to investigate that here as well.
Comment: Since the presence of protein in the structure of the flaxseed gum is responsible for its interfacial properties such as emulsifying ability and foaming ability, authors should have considered the effect of extraction temperature on the amount of the protein in the extracted gum. The content of the protein in the extracted gums at different temperatures should be measured and its influence on the gum functional properties should be discussed?
Response: We didn't investigate protein content in the gum at the time of the experiment.
- In Results and Discussion,
Comment: Line 217, a comma is needed after transmittance and before it.
Response: fixed
Comment: Line 227, a comma is needed after In addition and before the.
Response: fixed
Comment: Line 229, the title " Emulsification Properties" doesn’t seem to be for the mentioned discussion since the discussion is on the viscosity rather that emulsification properties.
Response: Changed the title: Viscosity During Storage
Comment: Line 293-319, Since the FG extracted at higher temperature possessed better surface-active properties, it is suggested that the effect of protein level in the two extracted gums will be discussed.
Response: Not available
Comment: In general, the discussion did not cover adequately the results of other researchers and it is essential to include and compare the findings of other studies, especially, new articles.
Response: We revised the discussion part and inserted 3 Refs as followings:
- Liu, J.; Shim, Y.Y.; Shen, J.; Wang, Y.; Ghosh, S.; Reaney, M.J.T. Variation of composition and functional properties of gum from six Canadian flaxseed (Linum usitatissimum L.) cultivars. Int. J. Food Sci. Tech. 2016, 51(10), 2313–2326.
- Chen, H.; Xu, S.; Wang, Z. Gelation properties of flaxseed gum. J. Food Eng. 2006, 77(2), 295–303.
- Moczkowska, M.; Karp, S.; Niu, Y.; Kurek, M.A. Enzymatic, enzymatic-ultrasonic and alkaline extraction of soluble dietary fibre from flaxseed–A physicochemical approach. Food hydrocolloids2019, 90, 105–112.
- Liu, X.; Han, R.; Yun, H.; Jung, K.; Jin, D.; Lee, B.; Min, T.; Jo, C. Effect of irradiation on foaming properties of egg white proteins. Poultr. Sci. 2009, 88(11), 2435–2441.
Comment: As a conclusion despite the interesting work, I recommend revision based on the mentioned above comments for publication in Foods. Manuscript should be rewritten to make it concise and jargon free.
Response: We have tried to improve the manuscript’s structure.

Round 2
Reviewer 1 Report
Dear authors
The content of manuscript improved but requires minor modifications. Please see the considerations below:
1 - The content of work still sounds incomplete regarding the discussion of results. Also, I am not a native English speaker but authors should revise the writing style. For instance:
- In the CONCLUSION section the authors are restricted only to compare the effects of temperatures. It is important to insert even in the conclusion for what your results would serve. One suggestion of rewriting: This work evaluated the functionality of flaxseed gum (FG) obtained with hot water extraction. The temperature increasing contributed to enhance the yield of FG, and the obtaining of product with improved emulsification properties and enhanced stability, which is of interest of food and non-food industries. Nevertheless, the product obtained presented dark coloration. It is expected that the results support further studies on the FG stability incorporated in food products.
- LINES 91-98. The authors should rewrite the argument that allowed the use of 70o The authors should improve the reason why the maximum of 98oC was used. One suggestion of rewriting: The obtaining of flaxseed gum consisted on the use of hot water extraction at 70 and 98oC. According to Xing et al and Cui et al the content of polysaccharide and protein in the gel increased with the temperature increasing and reached maximum at 70oC.
2 - In the INTRODUCTION the author should insert data regarding the production and consumption of flaxseed. For what the authors expect the results would serve?
3 - LINES 79-80 The authors use the concentrations of enzymes and microorganisms to discuss some results (LINES 309-312; 337-338). However, the authors did not measure these attributes. Which/How much of microorganisms are appropriate to maintain the stability of FG? Also, which enzymes must be present and at which activity?
4 LINES 272-274 – I did not understand. Is the word ´´mscreen´´ correct? Can the salt decrease the bond between protein and carbohydrates? Insert a li
5 LINE 300 - ´´IDEA´´ OR ´´IDEAL´´?
6 LINE 193 – What do you mean with ´´higher temperatures´´? Temperatures higher than 60oC?
Author Response
Manuscript ID foods-769047 entitled “Flaxseed (Linum usitatissimum L.) Gum Solution Functional Properties”
Thank you for your patience. We have revised our manuscript (Manuscript ID: foods-769047) according to reviewers’ comments/suggestions and would like to thank all reviewers for their critical feedback in making this manuscript more polished. We have listed reviewers’ comments and answered them in sequence. In addition to the changes made to the manuscript as recommended by the reviewers, we have also tried to improve the manuscript’s structure. We appreciate the reviewers’ thoughtful comments and critiques and hope this response addresses the overall quality of this manuscript for publication.
Responses to Reviewer 1 Comments and Suggestions:
The content of the manuscript improved but requires minor modifications. Please see the considerations below:
Comment 1 - The content of work still sounds incomplete regarding the discussion of results. Also, I am not a native English speaker but authors should revise the writing style. For instance:
In the CONCLUSION section the authors are restricted only to compare the effects of temperatures. It is important to insert even in the conclusion for what your results would serve. One suggestion of rewriting: This work evaluated the functionality of flaxseed gum (FG) obtained with hot water extraction. The temperature increasing contributed to enhance the yield of FG, and the obtaining of product with improved emulsification properties and enhanced stability, which is of interest of food and non-food industries. Nevertheless, the product obtained presented dark coloration. It is expected that the results support further studies on the FG stability incorporated in food products.
LINES 91-98. The authors should rewrite the argument that allowed the use of 70o The authors should improve the reason why the maximum of 98oC was used. One suggestion of rewriting: The obtaining of flaxseed gum consisted on the use of hot water extraction at 70 and 98oC. According to Xing et al and Cui et al the content of polysaccharide and protein in the gel increased with the temperature increasing and reached maximum at 70oC.
Response: Revised as recommended (Lines 97-101 and 361-366).
Comment 2: In the INTRODUCTION the author should insert data regarding the production and consumption of flaxseed. For what the authors expect the results would serve?
Response: We have added the following to the introduction “According to the latest annual flaxseed production records available in the statistical database of the Food and Agriculture Organization of the United Nations (FAOSTAT), Canada, China, Russia, the United States, Kazakhstan, India, and Ethiopia have produced the majority of flaxseed (2.41 million tonnes) over the last 20 years [3].”
Comment 3: LINES 79-80 The authors use the concentrations of enzymes and microorganisms to discuss some results (LINES 309-312; 337-338). However, the authors did not measure these attributes. Which/How much of microorganisms are appropriate to maintain the stability of FG? Also, which enzymes must be present and at which activity?
Response: Deleted the sentence (Lines 79-80) as your advice. Analysis of microorganism was not included in this manuscript; enzyme analysis was not performed.
Comment 4: LINES 272-274 – I did not understand. Is the word ´´mscreen´´ correct? Can the salt decrease the bond between protein and carbohydrates? Insert a li
Response: Corrected (Lines 274, 275).
Comment 5: LINE 300 - ´´IDEA´´ OR ´´IDEAL´´?
Response: Corrected (Line 302).
Comment 6: LINE 193 – What do you mean with ´´higher temperatures´´? Temperatures higher than 60oC?
Response: higher temperature means increased temperature gives greater recoveries of flaxseed gum (Line 195).

Reviewer 2 Report
The manuscript has been improved and it can be accepted in its present form.
Author Response
Thank you